# Multi-Region Microdialysis Imaging Platform Revealed Dorsal Raphe Nucleus Calcium Signaling and Serotonin Dynamics during Nociceptive Pain

**DOI:** 10.3390/ijms24076654

**Published:** 2023-04-03

**Authors:** Latiful Akbar, Virgil Christian Garcia Castillo, Joshua Philippe Olorocisimo, Yasumi Ohta, Mamiko Kawahara, Hironari Takehara, Makito Haruta, Hiroyuki Tashiro, Kiyotaka Sasagawa, Masahiro Ohsawa, Yasemin M. Akay, Metin Akay, Jun Ohta

**Affiliations:** 1Division of Materials Science, Graduate School of Science and Technology, Nara Institute of Science and Technology, Ikoma 630-0101, Japan; 2Department of Health Sciences, Faculty of Medical Sciences, Kyushu University, Fukuoka 819-0395, Japan; 3Department of Neuropharmacology, Faculty of Pharmaceutical Sciences, Nagoya City University, Nagoya 467-8601, Japan; 4Biomedical Engineering Department, University of Houston, 3517 Cullen Blvd, Houston, TX 77204, USA

**Keywords:** calcium imaging, DRN, nociception, pain, serotonin, micro-imaging device, microdialysis

## Abstract

In this research, we combined our ultralight micro-imaging device for calcium imaging with microdialysis to simultaneously visualize neural activity in the dorsal raphe nucleus (DRN) and measure serotonin release in the central nucleus of the amygdala (CeA) and the anterior cingulate cortex (ACC). Using this platform, we observed brain activity following nociception induced by formalin injection in the mouse’s hind paw. Our device showed that DRN fluorescence intensity increased after formalin injection, and the increase was highly correlated with the elevation in serotonin release in both the CeA and ACC. The increase in calcium fluorescence intensity occurred during the acute and inflammatory phases, which suggests the biphasic response of nociceptive pain. Furthermore, we found that the increase in fluorescence intensity was positively correlated with mouse licking behavior. Lastly, we compared the laterality of pain stimulation and found that DRN fluorescence activity was higher for contralateral stimulation. Microdialysis showed that CeA serotonin concentration increased only after contralateral stimulation, while ACC serotonin release responded bilaterally. In conclusion, our study not only revealed the inter-regional serotonergic connection among the DRN, the CeA, and the ACC, but also demonstrated that our device is feasible for multi-site implantation in conjunction with a microdialysis system, allowing the simultaneous multi-modal observation of different regions in the brain.

## 1. Introduction

Pain is a natural protective response essential to avoid potential tissue damage or injury [1]. However, when pain persists longer, it can lead to suffering, transitioning from acute to chronic pain [2], which costs a clinical and economic burden. Chronic pain is one of the leading causes of disability and also the most frequent reason for seeking medical treatment [3]. For more than a century, researchers have identified the pain matrix in the brain [4,5], involving multidimensional aspects, such as sensory-discriminative and affective-emotional components [6], which can differ based on the intensity of an unpleasant feeling. This painful sensation is detected by the nociceptors, then transmitted through the complex processing in the central nervous system (CNS) [7]. To this day, understanding the principles and mechanisms that govern neural connectivity during the encoding and decoding of pain signals is still one of the major challenges in pain research. Given the importance of multiple brain areas in modulating and regulating pain, observing brain activities in various locations at the same time is necessary for elucidating this phenomenon. Therefore, in this paper, we integrated calcium imaging and microdialysis to concurrently observe neural activity and measure neurotransmitter release in freely moving mice during nociception.

Visualizing neural activity is crucial in understanding the biological process in the brain. As the fundamental mechanism that underlies brain function and behavior remains elusive, rigorous work is being conducted using new methodologies to shed light on the connections and functional properties of the nervous system. Therefore, optical imaging has been developed as a promising technique since it offers a real-time recording of neural activity in vivo. Incorporating this device with a genetically encoded calcium indicator (GECI), such as GCaMP [8], enables us to perform fluorescence imaging in a freely behaving condition [9,10,11,12]. Our lab has developed a thin and lightweight fluorescence imaging device (0.02 g) using a complementary metal-oxide semiconductor (CMOS) image sensor with µLED-based excitation light [13]. This design can avoid bulky optical compartments while retaining the function of observing neurons. It enables simultaneous fluorescence measurement in multiple regions and can be used with other modalities.

In previous studies, we have demonstrated the utilization of our novel imaging device for observing deep brain regions [14,15]. The core work of this study is to apply our imaging device for pain circuitry and combine it with microdialysis. This allows us to measure two proxies of brain activity: (1) calcium imaging fluorescence, and (2) neurotransmitter release. We focus the study on serotonin (5-hydroxytryptamine or 5-HT), a neurotransmitter that has a well-known correlation with the regulation of pain [16]. The depletion of serotonin can increase pain sensitivity [17]. Additionally, the serotonergic pathway is also currently used as a potential therapeutic target against some chronic pain problems.

We implanted the device in the dorsal raphe nucleus (DRN), which contains more than half of the serotonergic cell body in the brain [18]. The DRN is located in the brain stem—specifically, in the ventral part of periaqueductal grey (PAG). Serotonin in the DRN is ascendingly projected to various brain regions, including the central nucleus of the amygdala (CeA) and the anterior cingulate cortex (ACC). The CeA in the limbic system is densely innervated by DRN serotonin neurons, is responsible for emotion [19], and is prominently perceived as a center for nociceptive-specific information. It plays a key role in pain [20], anxiety [21], and depressive disorder [22]. The ACC is also an essential area in the cortex that is actively involved in pain processing and chronification [23,24]. Therefore, in this research, we implanted microdialysis probes in the CeA and the ACC (Figure 1). We injected formalin into the mid plantar of the mouse’s hind paw, which is a widely used model for nociceptive pain [25,26].

The aim of this study is to demonstrate the utility of our imaging device in monitoring calcium activity and conducting multiple implantations in conjunction with microdialysis without hindering the mouse’s movement. We observe the dynamics of the serotonergic system in the higher brain region. We also identify the brain response toward the laterality of pain stimulation. With this microdialysis imaging platform, we are able to collect the fluorescence data and serotonin release from various locations simultaneously, which provides a better understanding of the multi-region interaction and behavior–brain activity relationship during nociception.

## 2. Results

### 2.1. Formalin-Induced Pain Increased Calcium Fluorescence Intensity in the DRN

We observed fluorescence activity in the DRN of G-CaMP6 transgenic mice, strain FVB-Tg (Thy1-GCaMP6)5Shi [27], using our implantable imaging device. This mouse strain produces fluorescence activity in all activated neurons. All the devices were implanted in the left hemisphere. Mice were injected with formalin (5%) into the right hind paw to induce acute pain condition, and the other group was injected with phosphate-buffered saline (PBS) as a control (Figure 2a,b). The average fluorescence activity every 5 min after the injection was successfully imaged (Figure 2c). Distinct and brighter areas were perceivable in the formalin group, particularly in the ventral part of the DRN, while the PBS group showed relatively the same uniform pattern throughout the measurement.

We then selected the regions of interest (ROIs) (Figure 3a) in every mouse (*n* = 3 mice for each group) using a custom-built algorithm based on adaptive binarization. Although it appears that there are no features in the average image of the PBS group, there are still ROIs that are selected by the algorithm. The algorithm checks for features in every frame of the video using adaptive binarization, hence, even low-intensity fluorescent forms can be detected as long as they were present at some point in the video.

Fluorescence activity in the DRN was observed, in which the intensity increased after the mice were injected with formalin, but not in the PBS group. This rapid change was especially apparent immediately after formalin injection, as shown in Figure 3b. The representative color plot in this figure is composed of the selected ROIs. This result is indicative of the involvement of the DRN during acute pain.

The fluorescence changes (Δ*F*/*F*_0_) averaged to a 5 Hz resolution for 1 h after the injection is shown in Figure 3c. Fluorescence changes were discretized to 5-min intervals, and we found statistically significant differences between the formalin (*n* = 63 ROIs, 3 mice) and PBS groups (*n* = 76 ROIs, 3 mice) that coincide with the acute and inflammatory phases of formalin-induced nociception (Figure 3d). There was a drastic enhancement in fluorescence intensity at around 10 min post-injection (*p* < 0.001), which may indicate the acute phase. Statistical differences in fluorescence amplitude were also found at 30 min (*p* < 0.001), 35 min (*p* < 0.01), and 40 min (*p* < 0.01) post-injection. These activities may reflect the inflammatory phase. These results demonstrated that the device is capable of visualizing the biphasic nociceptive responses in the DRN.

We also observed the suppression activity in the DRN by injecting lidocaine (FUJIFILM Wako Pure Chemical Co., Ltd., Osaka, Japan) into the intrathecal space 10 min prior to formalin injection. Lidocaine is well-known as a local anesthesia that can block neural conduction by inhibiting voltage-gated sodium channels (Na+). The fluorescence activity of the Lidocaine group was relatively lower compared to the formalin-only group (Appendix A). Immunostaining results also corroborate the results of our devices. We confirmed that more neurons were activated as labeled by cFos in the formalin group compared to the PBS group in the DRN (Appendix A).

### 2.2. Formalin-Induced Pain Increased Serotonin Levels in the CeA and the ACC

Concurrently with calcium image recording, we measured serotonin levels in the CeA and the ACC (Figure 4a) using a microdialysis probe implanted in the left side of the respective brain region coupled with high-performance liquid chromatography (HPLC). Serotonin concentration increased in the CeA (*p* < 0.05) (Figure 4b) and the ACC (*p* < 0.01) after the formalin injection, but not after the PBS injection (Figure 4c).

The highest serotonin concentration in the CeA and the ACC was observed in the first sample collection after the injection (15 min per sample), which then gradually declined. Due to the temporal resolution of microdialysis, it is difficult to differentiate the acute phase from the inflammatory phase. Additionally, suppression of serotonin release was observed after lidocaine injection into the intrathecal space (Appendix A). This parallels the results we found in the DRN imaging. We further validated the elevation of serotonin release in the CeA and the ACC using immunostaining, which showed more activity in cFos-positive cells after formalin injection compared to PBS injection (Appendix A).

### 2.3. Formalin-Induced Pain Generated Lateral Responses in the DRN, CeA, and ACC

In this study, we implanted three devices in the left hemisphere. Then, we injected a solution (formalin or PBS) either into the right (contralateral) or the left (ipsilateral) hind paw (Figure 5a). We tested the response of the DRN, the CeA, and the ACC after formalin injection. A statistically significant difference in the fluorescence intensity of the DRN was found between contralateral and ipsilateral stimulation at certain time points (Figure 5b). These time points were in the acute phase at 10 min after the injection (*p* < 0.01), and during the inflammatory phase at 30 min (*p* < 0.001) and 40 min (*p* < 0.05). Serotonin release in the CeA was also statistically different between contralateral and ipsilateral (*p* < 0.05) (Figure 5c). Serotonin concentration in the CeA increased compared to pre-stimulation only after contralateral injection. In contrast, serotonin release before and after the injection in ipsilateral stimulation was not statistically different (Figure 5e).

On the other hand, the ACC had a different reaction. Serotonin release in the ACC showed a higher serotonin concentration in contralateral compared to ipsilateral (*p* < 0.05) (Figure 5d). However, if we compared pre- and post-stimulation, both serotonin in contralateral and ipsilateral were enhanced after formalin injection (Figure 5f). This implies that the ACC responds bilaterally toward formalin stimulation, while the CeA responds unilaterally. The distinction in the extracellular serotonin release between the CeA and the ACC may indicate the different mechanisms of these two brain regions in response to formalin injection.

### 2.4. Calcium Fluorescence Intensity and Serotonin Release Were Correlated after Formalin Injection

In order to correlate fluorescence activity and serotonin release, we counted the number of peaks in calcium imaging traces. Peaks were defined as intensities higher than two deviations from the mean of the bandpass (0.17–4.5 Hz) filtered data. A representative peak raster plot for formalin and PBS stimulation is shown in Figure 6.

We found a strong positive correlation between the number of peaks in the fluorescence activity in the DRN and the serotonin release in the CeA (*ρ* = 0.9) for the formalin group using Spearman’s rank correlation. The same monotonic relationship was also found for serotonin release in the ACC (*ρ* = 0.8). In contrast, in the PBS group, serotonin concentration in the ACC and fluorescence activity showed a weak correlation (*ρ* = 0.3), while the correlation for serotonin activity in the CeA was negligible (*ρ* = 0.1) (Figure 7).

### 2.5. Calcium Fluorescence Intensity and Licking Behavior Were Correlated after Formalin Injection

The formalin test is widely used as an acute inflammatory pain model. During the nociception, the mice typically demonstrate pain-related behavior such as licking and biting toward the injected paw. It is well-established that the injection of formalin in the plantar of a mouse’s hind paw can generate biphasic behavior responses. We counted the duration of licking behavior every 5 min (as shown with the black line in Figure 8). The licking behavior sharply increased after 5 min of formalin injection followed by a steep decline. Then it moderately increased again after 30 min and started fluctuating. This shows the biphasic response during licking behavior. The behavior assay is important as it can reflect activity in the brain.

We then assessed the correlation between the peaks of DRN fluorescence with mouse licking behavior (Figure 8). Using Spearman’s rank correlation test, we found that the peaks of the fluorescence activity in the formalin group were strongly correlated with licking behavior (*ρ* = 0.74). On the other hand, the PBS group showed a weaker correlation (*ρ* = 0.46).

## 3. Discussion

### 3.1. CMOS Implantable Device for Probing Neural Activity

Calcium imaging is a powerful technique that is widely used in neuroscience for indirectly investigating neural activity. It measures the intracellular calcium concentration as a proxy for action potential [28]. In this study, we demonstrated the utility of our novel imaging device to observe fluorescence activity from GCaMP-expressing cells in the DRN of freely behaving mice. Our imaging results demonstrated that the device was able to detect a difference between the pain and the control groups. The increase in calcium signal after formalin injection was perceivable, while the activity in PBS-injected mice did not increase.

The ability of this device to detect the fluorescence activity in the specific brain region makes use of the basic principles of in vivo calcium imaging. The changes in fluorescence intensity captured by the CMOS image sensor indicate the calcium concentration changes during neural activation. Considering the decay time of GCaMP [8], an image acquisition rate of 10 fps is sufficient for observing the changes in calcium signals. However, unlike lens-based systems such as devices using GRIN lenses, individual neurons are difficult to observe with our device due to the lack of an optical system. Nevertheless, fluorescent forms or shapes can be assumed to be neuronal populations based on size. The size of these forms is affected by light scattering, but the contribution of this has been experimentally determined [29]. Increasing the sensitivity and resolution of the imaging device can also be further improved by using hybrid optical filters [30,31], narrow-band light sources [32,33], and improved pixel design [34,35].

Despite the limitations of our lensless device, there are plenty of advantages. Weighing only 0.02 g, our device is the lightest among other current commercially available imaging devices used for freely moving mice such as the most current 1P miniscope (e.g., Miniscope v4, 2.6 g) and 2P miniscope (e.g., Mini2P, 3 g) [36]. Since the device is ultralight and small (200 µm in thickness), (1) implantation can be less invasive due to less tissue damage, (2) implantation is easier, (3) a more natural movement can be observed under freely moving conditions due to less hindrance, and, finally, (4) multi-region observation can be conducted. The naturalness of movement is a crucial consideration when studying the relationship between brain activity and behavior. Furthermore, multi-region observation is difficult, if at all possible, using the above-mentioned other devices. In terms of advantages, our device is comparable to fiber photometry, but the electrical wires we use in our system are more flexible than optic fibers. Moreover, while fiber photometry is good at observing in the horizontal plane, our imaging device can visualize the vertical plane and has a wider field of view, allowing multiple-region imaging with only one implanted device so that it can observe the difference in the roles of the regions.

### 3.2. Biphasic Response in the DRN Imaging during Nociception

Calcium imaging has been extensively used for studying neural activity under various physiological conditions in the DRN, such as the reward system and social interaction [37,38], threat-induced locomotion [39], and emotionally salient behavior [40]. However, little research is focused on employing this technique to investigate acute pain and hyperalgesia in the DRN, even though this region is crucial in pain modulation [41]. Multiple studies have used the formalin–pain model in assessing analgesic drugs or other preclinical research [42], yet calcium imaging in the DRN during formalin stimulation remains unexplored. In this study, we used our imaging device to observe neuronal activity in the DRN. We found that the fluorescence activity increased and the biphasic pain response appeared after formalin injection, suggesting the contribution of DRN neurons in nociceptive signal processing. This further builds upon the results of a previous study using our CMOS imaging device to monitor nociception [14].

The biphasic response is the typical reaction evoked after formalin injection. It generally shows two separate peaks that represent acute and inflammatory stages, with variation in the duration of each phase. The acute phase usually occurs at 5 min [43,44] or up to 10 min [45,46,47], while the inflammatory phase varies and can occur from 20–50 min after injection [48,49,50]. This response has been shown in numerous studies through behavior [48,51,52,53]. However, this has not been demonstrated as much through calcium imaging. Previously, the biphasic response was shown in the dorsal root ganglion (DRG) during nociception [54]; however, there has been a lack of demonstration of this phenomenon in the brain.

Our imaging results in the DRN demonstrated that the biphasic response still happens at the supraspinal level. The highest peak of fluorescence intensity in the initial stage (10 min post-injection) was attributed to the acute phase that might occur due to the direct activation of the nociceptive afferent neurons [46], while in the late stage, the increased fluorescence activity that appeared around 20–40 min was most likely caused by the inflammatory reaction and the increase in central sensitization in the dorsal horn [50]. In this phase, fluorescence activity tends to increase several times, but with a lower intensity due to different timings per mouse. Similarly, individual neurons in the DRG showed a strong response and consistent timing in the acute phase but a weaker response and variable timing in the inflammatory stage [54]. This may also explain the fluctuation of nociceptive signals in the DRN during the inflammatory stage, given the heterogeneous structure and functionality of DRN at the cellular and molecular levels [18,55].

The majority of the neurons in the DRN are serotonergic [56,57]. Among them, over 2000 DRN serotonergic neurons orchestrate serotonin release throughout the brain [40]. These neurons are involved in various physiological processes in the brain, including pain. The ascending serotonergic pathway from the DRN to the forebrain areas and the limbic system controls the attentional processing of nociception [58]. The alteration of serotonin in the DRN can lead to different pain sensitivity [59]. In addition, the exogenous administration of serotonin can generate an analgesic effect, and one of the current potential treatments for pain is by using selective serotonin reuptake inhibitors (SSRIs) to increase serotonin [60].

An increase in the calcium activity of the DRN might affect serotonin release to other brain regions. In this study, however, we used G-CaMP6, which is not specific to the serotonergic neurons. Using cell type-specific mice such as SERT-Cre transgenic lines can help determine whether the calcium activity is mainly caused by serotonergic or non-serotonergic neurons. Moreover, previous studies revealed the dual function of serotonin in facilitatory and inhibitory pain signaling, depending on the location and state of the 5HT receptor subtypes [61]; thus, using even more specific 5HT receptor-based calcium reporters would be useful for future studies. Nonetheless, to further explore the role of DRN calcium activity in serotonin release in other brain regions, we employed microdialysis measurement in the CeA and ACC.

### 3.3. Microdialysis Imaging Platform for Investigating Pain in the Central Nervous System

Previous studies have applied two microdialysis probes in rat brain to investigate the antibody pharmacokinetics in the two brain regions [62]. Implanting the two probes allows us to measure neurotransmitters from different locations or observe the same region in the two different hemispheres simultaneously. In the present study, we introduce a platform consisting of the integration of two microdialysis probes, with one micro-imaging device. According to our extensive literature search, this is the first report on a three-device implantation in mouse brain for the simultaneous recording of in vivo calcium imaging and neurotransmitter release under a freely moving condition. This platform is beneficial for observing neural activities and the release of extracellular neurotransmitters from various locations at the same time, providing us an insight into how the neural network is orchestrated during pain processing in the central nervous system. Using this platform, we were able to investigate the serotonergic system in the pain circuits. We found an increase in fluorescence activity in the DRN after formalin stimulation, which was in line with the increase in serotonin concentration in the CeA and the ACC.

In the microdialysis result, the peak serotonin release in response to formalin injection can be seen in the initial stage, the first 15 min after the stimulation. Previous studies have been conducted to observe serotonin release in the amygdala using in vivo microdialysis with various stressors, for example, by injecting a corticotrophin-releasing factor [63], d-Phe-CRF [64], amphetamine (stress inducer) [65], conditioned fear [66], and noxious stimuli [67]. In this study, the increase in serotonin levels after formalin injection supports the important aspect of the amygdala in pain modulation. The amygdala is known to have a key role in fear, anxiety, and depression [21,22]. It is also actively involved as the center for an affective–emotional component of pain [68]. Serotonin can influence emotion via the modulation of amygdala function, which is usually related to aversive behavior [69]. Moreover, the central amygdala, particularly the lateral capsular subsection, receives specific information about the nociceptive signals [70] that are innervated by the serotonin neurons from the DRN. Given the importance of the DRN–amygdala projection in the serotonergic system, it is believed that the increase in firing rates in the DRN will elevate the serotonin levels in the amygdala [69].

Serotonin concentration was also found to rise after formalin injection in the ACC. This suggests the involvement of the ACC in pain modulation through serotonin signaling. Indeed, the ACC is innervated by serotonergic terminals, and some serotonin receptors, such as 5HT1A and 5HT7, are well-documented to contribute to pain processing. In addition, extensive studies from humans and rodents have shown the fundamental roles of the ACC in acute and chronic pain [71]. For instance, serotonin in the ACC contributes to relieving pain and anxiety through synaptic plasticity triggered by regular aerobic exercise [72].

The advantages of using this microdialysis imaging platform are that we can observe the activity in the different locations simultaneously, and thus the interaction among different regions can be observed. Therefore, we assessed the correlation between the targeted regions in the studies. We found a high positive correlation between DRN calcium imaging and the serotonin release in the CeA and the ACC, which is in line with the typical projections of the serotonergic system from the DRN to the higher brain regions. These two regions, the CeA and the ACC, are innervated by serotonin neurons from the DRN. A previous study using optogenetics also mentioned a strong connection between the DRN serotonergic neurons with the ACC [73] and the amygdala [74,75].

This microdialysis imaging platform can be used to understand brain activity, which is a dynamic system. Pain is not isolated from one region, but rather a manifestation from multiple regions in the brain, combining the complex signaling pathway from somatosensory, emotional, and cognitive aspects that finally form pain perception. This platform should be further implemented to explore different neural circuits in other psychiatric disorders.

### 3.4. Laterality in Nociceptive Pain Processing

Many studies have discussed that pain exhibits lateralization [76]; notably, neuroimaging studies have revealed that the widespread pain matrix across the brain may have distinct lateralized functionality [77,78]. The term lateralization in most of the studies is commonly referred to as hemispheric lateralization [79,80], owing to the different brain processing in the left and the right hemispheres. Particularly, one hemisphere would have a lower pain threshold than the others depending on the site of injury. However, in this study, we assessed the laterality of pain processing of the same brain hemisphere, as a response to unilateral formalin stimulation. It is interesting to see that the different brain regions may behave in a distinct fashion; for example, in this study, we found different responses of serotonin release of contralateral and ipsilateral stimulation in the CeA and the ACC. The CeA shows a laterality effect, which is supported by a previous study [67]. This study suggested that the noxious stimuli triggered the increase in serotonin in the CeA contralaterally, yet innocuous stimuli decreased serotonin release in both the contralateral and ipsilateral. Therefore, it is possible that the laterality effect in this study may be caused by a different mechanism, which depends on the stimulation modality and the serotonin signaling pathway in the pain circuits.

The laterality effect in the DRN and CeA may be due to the signal transmission in the pain pathway. In general, when the noxious stimuli exceed the pain threshold, the nociceptive signals that are conveyed and transduced by the specialized peripheral neurons will travel through primary afferent neuron fibers (Aδ and C fibers) into the DRG [81]. Then, this substantial nociceptive signal will terminate in the lamina I of the spinal horn [82]. The projection from lamina I is approximately 85% contralateral to the higher brain regions, such as LPb and PAG [83]. The signal decussates in the opposite direction via the midline crossing of the ascending tract before it reaches the brain [84]. This may explain the response of calcium signaling in the DRN and the unilateral response in CeA serotonin release. However, in higher cortical areas, this laterality may be reduced due to the interhemispheric connection through the corpus callosum [85].

As shown in our results, serotonin release in the ACC increased in both contralateral and ipsilateral stimulation of formalin. This result is in accordance with the previous study using calcium imaging of the ACC in awake, head-restrained mice, which demonstrated that noxious stimulation (10 g von Frey) activated the ACC neurons in both contralateral and ipsilateral conditions [86]. Similar to our study, there was no clear predominance of contralateral and ipsilateral response in the ACC. This strengthens the idea that the cytoarchitecture of the brain permits bilateral interactions of neurons between the right and left hemispheres of the ACC via cross-callosal projections [87].

Future studies can confirm the mechanism of pain in higher brain areas. By using our multi-region observation technique together with pharmacological or optogenetic interventions, investigating the neural circuit of nociception pain perception can be performed. In addition, since our study was focused on acute pain, further research can be conducted to assess formalin-induced secondary allodynia and hyperalgesia. This can help us understand the transition from acute to long-term hyperalgesia, and eventually chronic pain.

## 4. Materials and Methods

### 4.1. Imaging Device Fabrication

Our lab has developed an implantable micro-imaging device for rodents [13,14,15]. The needle type of this CMOS-based device allows us to observe neural activity in the deep brain region. The implantable imaging device structure consists of a CMOS image sensor and µLED embedded in a flexible printed circuit (FPC) (Taiyo Industrial Co. Ltd., Wakayama, Japan) (Figure 9a). We used a CMOS image sensor containing 40 × 120 pixels (7.5 μm per pixel). The imaging device has a long vertical view, allowing us to observe multiple layers at the same time. To excite GCaMP-expressing cells, we used a µLED with an emission wavelength of 473 nm (ES-VEBCM12A, Epistar, Hsinchu, Taiwan). The fabrication of this imaging device was performed through the following processes: (i) the attachment of the image sensor and µLED to the FPC, (ii) filter application to the image sensor, (iii) wire bonding and red resist application, and (iv) epoxy encapsulation and parylene coating (Figure 9b).

The absorption filter was made by mixing Valifast yellow 3150 dye (Orient Chemical Industries, Osaka, Japan) with cyclopentanone and resin (Norland Optical Adhesive 63, Cranbury, NJ, USA). The solution was spin-coated (3 s slope, 5 sec at 500 rpm, 5 s slope, 20 s at 2000 rpm, and 5 s slope to 0 rpm) on a silicone-coated cover slip, then it was cured by heating at 90 °C for 30 min and exposed to UV for 30 s. The filter was cut in accordance with the imaging sensor size using a YAG laser machine (Callisto VL-C30RS-GV, TNS Systems). The filter was applied to the surface of the image sensor and baked for 120 min (120 °C) under vacuum (vacuum oven, AVO-250NS, ETTAS). The absorption filter was placed on the surface of the CMOS chip so that only emission light from the activated GCAMP6-expressing cells would reach the imaging area. Red resist resin (ST-3000L, Fujifilm, Tokyo, Japan) was added to the edges of the chip to prevent light from the side reaching the sensor. The electrical pads of the CMOS chip and µLED were connected to the FPC circuitry via aluminum wire (TANAKA Holdings Co., Ltd., Tokyo, Japan) using a wire bonder machine (7400C-79, West Bond, Anaheim, CA, USA). Epoxy resin was applied (Z-1; Nissin resin, Yokohama, Japan) to protect those wires. The imaging device was removed from the FPC, and sub-sequentially, a completed device was coated with Parylene-C (5 g of dichloro-c-cyclophane) using a perylene coating chamber (PDS 2010, Specialty Coating Systems). The thickness of this coating was 2.5 µm, which added a biocompatibility feature to the device.

### 4.2. Animal Surgery and Device Implantation

In this research, we used G-CaMP6 transgenic mice, strain FVB-Tg (Thy1-GCaMP6)5Shi (RBRC09452, RIKEN) [27] to measure the activity of the DRN using fluorescence calcium imaging. Before the surgery, the mouse was anesthetized using a mix of three anesthetic agents (0.3 mg/kg of medetomidine, 4.0 mg/kg of midazolam, and 5.0 mg/kg of butorphanol) administered intraperitoneally [88]. The scalp hair was trimmed after being fully sedated. The mouse was mounted on a stereotaxic instrument (SR-6M, Narishige, Tokyo, Japan), and the head was secured with an ear bar. A heating pad was placed to maintain a safe temperature. Chlorhexidine gluconate (antiseptic) was applied before the surgery. Horizontal and rostral incisions were made, and the skin was removed to expose the cranial surface. To prevent drying of the surgical site, PBS (Fujifilm Wako, Osaka, Japan) was added. Bregma was marked, and its position was confirmed to be on the same level as lambda. The implanted sites were determined using the Paxinos Mouse Brain Atlas (Franklin and Paxinos 2008). The stereotaxic coordinates were as follows: the DRN (anteroposterior (AP): −5.56 mm, mediolateral (ML): 0.35, and dorsoventral (DV): 3.31 mm) tilted 25° anteriorly), the CeA (AP: −0.8 mm, ML: 3.35 mm, top guide 3 mm), and the ACC (AP: 1, ML: 0.3, top guide 0.6 mm, tilted 10° posteriorly). Since our device offers flexibility because it is light and small, it was possible to perform tilted implantation. The tilted angle was implemented in DRN to prevent hitting large blood vessels during the implantation, while the tilted angle of the microdialysis probe in the ACC was implemented to avoid the collision between the probes of the ACC and the CeA. Sealant (Kwik-Cast) and dental cement (Super-Bond kit, Sun Medical) was used to cover and protect the exposed brain surface. The mouse was returned to the cage and allowed to recover before undergoing an imaging and microdialysis experiment.

### 4.3. Imaging and Microdialysis Experiment

Imaging and microdialysis were performed the next day after surgery. Before conducting this experiment, the previously implanted dummy probe was replaced by the dialysis probe. This probe contains a semipermeable membrane (1 mm), which allows low-molecular-weight solutes, such as neurotransmitters, from the extracellular space in the brain to be collected. Ringer’s solution (NaCl 8.6 g/L, KCl 0.3 g/L, CaCl_2_ 0.25 g/L) was perfused through the target site with a flow rate of 1 µL/min. The sample were collected every 15 min. The dialysate (10 µL) was injected into a high-performance liquid chromatographic column-electrochemical detector (HPLC-ECD) to measure serotonin concentration. The calibration curve was made by injecting 0.50 pg/µL, 1.0 pg/µL, and 10.0 pg/µL of standard serotonin solution into the HPLC. The standard serotonin solution was prepared by mixing 100 ng/μL serotonin with 0.1 mol/L acetic acids containing 100 mg/L EDTA/2Na. For the calibration, each solution was diluted with 0.1 mmol/L EDTA·2Na (37 mg/L) containing 0.1 mol/L phosphate buffer (pH 3.5).

The imaging device was connected to a power supply (6146 DC Voltage Current Source, ADC Corporation, Saitama, Japan). The imaging data were obtained at 10 frame rates per second (fps). The mouse was allowed to habituate 2 h before the pain stimulation, and this period was used as the baseline. The mouse was monitored and recorded using a bandicam. In this imaging experiment, we excited the G-CaMP6 neurons in the DRN using µLED, and the CMOS image sensor captured the fluorescence emission from the G-CaMP6-expressing cells. Using this microdialysis imaging system, serotonin release was observed in the ACC and the CeA while simultaneously neural activity was observed in the DRN following acute noxious stimulation for 1 h. Figure 10 illustrates the principle of the microdialysis imaging system.

### 4.4. Pain Stimulation

Paraformaldehyde was used as a pain group, and PBS as a control. Paraformaldehyde was injected into the right (contralateral to the implanted device) or the left side (ipsilateral to the implanted device) of the mouse’s mid-plantar hind paw to produce formalin-induced nociceptive pain, while PBS was injected in the control group. The formalin test is widely used to assess pain-associated behavioral responses [25,89]. Brain imaging and mouse licking behavior were observed for 1 h under contralateral and ipsilateral conditions.

### 4.5. Brain Slicing and Immunostaining

After completing the experiment, mice were sacrificed, and the brain was extracted with the perfusion-fixation technique. The brain was fixed using 4% paraformaldehyde (PFA) for 4 h and moved to a mixed PBS and 0.1% sodium azide (Wako Pure Chemical Industries, Inc., Osaka, Japan). Coronal brain slicing (40 µm thickness) was performed using a vibratome (Linear Slicer PRO7, Dosaka, Kyoto, Japan) to confirm the implantation of the imaging device and microdialysis probes on the targeted sites (Appendix A). Immunostaining was also conducted to identify the activated neurons. The brain slices were placed in the glass slides and dripped with 50 μL of blocking solution (4% FBS/0.1% Triton X-100/0.05% NaN3/PBS), then incubated overnight at 4 °C. The primary antibody (50 μL), which diluted the blocking solution, was poured onto brain slices for 1 h at room temperature, following the secondary antibody in the same fashion. Primary antibodies consisted of anti-NeuN antibody (Host: Mouse, Monoclonal Antibody, ab104224, Abcam) for staining the neurons in the CeA and the ACC, anti-Tryptophan hydroxylase 2 (TPH2) antibody (Host: Rabbit, Polyclonal Antibody, ab125877, Abcam, Tokyo, Japan) to identify the serotonergic neurons in the DRN, and anti-c-Fos antibody (Host: Mouse, Monoclonal Antibody, ab208942, Abcam) was used to observe the activated neurons after the stimulation. Anti-mouse IgG Alexa Fluor 488, and goat anti-rabbit IgG Alexa Fluor 568 were used as the secondary antibody. The staining results were observed under an inverted microscope (DMI6000, Leica Microsystems, Tokyo, Japan).

### 4.6. Data Analysis

A custom-made CMOS imaging system called CIS_NAIST was used to collect data from our imaging devices. Each 5-h recording session contained two stimulation events and was trimmed down to two 90-min videos comprising 10 to 40 min before stimulation and 60 min after stimulation. The recording 10 min before the stimulation up to the point of the stimulation contains high intensity values due to external light sources required to inject the chemical for stimulation. Thus, this part of the video was intentionally removed. Each pixel was then normalized to the average fluorescence value of the baseline, which is defined as 15 min before the stimulation event in the 90-min video. Mathematically, each pixel value is converted to %Δ*F*/*F*_0_ defined by the following equation:$$\%\Delta F/F_{0}=\frac{F-F_{0}}{F_{0}}\times 100$$
where *F* is the fluorescence intensity of the pixel and *F*_0_ is the baseline fluorescence.

Region Of Interest (ROI) selection was conducted using an algorithm consisting of a series of adaptive binarization and morphological transformations (Figure 11). All operations were conducted with an image processing library in Python version 3.10.6 (http://www.python.org/ (accessed on 16 January 2023) called scikit-image [90]. Each frame after stimulation was locally binarized with a gaussian method using a kernel size of 9 and a sigma value of 1.5. The binarized frame was then cleaned by morphological binary opening with a footprint size of 2 and morphological area opening with an area size of 15. The binarized frames were then averaged and binarized again with a kernel size of 9 and a sigma value of 2.5. The binarized average frame was then cleaned with the same morphological operations in the same order with a footprint size of 3 and an area size of 15. Each form in the resulting image was used as an ROI.

The average fluorescence of each ROI was plotted against time and normalized by division to the baseline fluorescence of the ROI. The normalized *%*Δ*F*/*F*_0_ time series of each ROI for each mouse of the same treatment group were then pooled before conducting statistical tests. Although the frame rate for each experiment was approximately 10 Hz, each experiment still had different frame rates (9.73–10.68 Hz). Thus, just for the visualization, each *%*Δ*F*/*F*_0_ time series was averaged to a 5 Hz time resolution and a 0.02 Hz lowpass filter was applied so that means and standard errors for each time point could be calculated. Whenever appropriate during statistical comparisons, each *%*Δ*F*/*F*_0_ time series was binned to certain time intervals and averaged per bin to match the time resolution of the corresponding data it was being compared to. For licking data, the *%*Δ*F*/*F*_0_ time series were discretized to 5-min intervals, while for microdialysis data, they were discretized to 15-min intervals. All statistical analyses were conducted with the stats subpackage of the SciPy library of python.

For the microdialysis data, the baseline used was the sample collected 15 to 0 min before the stimulation. For the comparison between pre- and post-stimulation release of serotonin, the sample collected 15 to 0 min before and 0 to 15 min after stimulation was used, respectively. *t*-test was performed to check for the statistical differences between the two.

## 5. Conclusions

In this study, we developed a multi-region microdialysis imaging system for investigating neural activity during nociception. Using this system, we found the biphasic response in the DRN calcium imaging during nociception, which was correlated to the licking behavior of formalin-induced pain. The increase in the DRN activity also corresponded with the elevation of serotonin release in the CeA and the ACC, suggesting the involvement of the serotonergic system in pain modulation. Furthermore, we evaluated the laterality of brain response to pain stimulation. We found that the activity in the DRN and the CeA was higher in contralateral stimulation, while the activity in the ACC was the same for both stimulation conditions. Finally, we showed that the DRN fluorescence activity was highly correlated to serotonin release in the CeA and the ACC. This research demonstrated the utility of our microdialysis imaging platform to observe the multi-region interaction of the pain circuit in the brain.

## Figures and Tables

**Figure 1 ijms-24-06654-f001:**
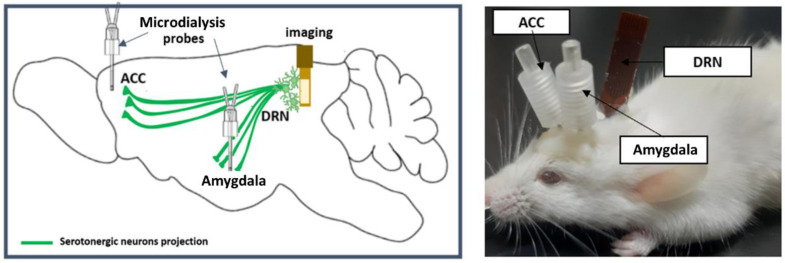
Illustration of the microdialysis imaging platform. (**Left**): serotonergic projection from the DRN to the CeA and the ACC. (**Right**): imaging device implantation in the left DRN and the microdialysis probes in the left CeA and the ACC.

**Figure 2 ijms-24-06654-f002:**
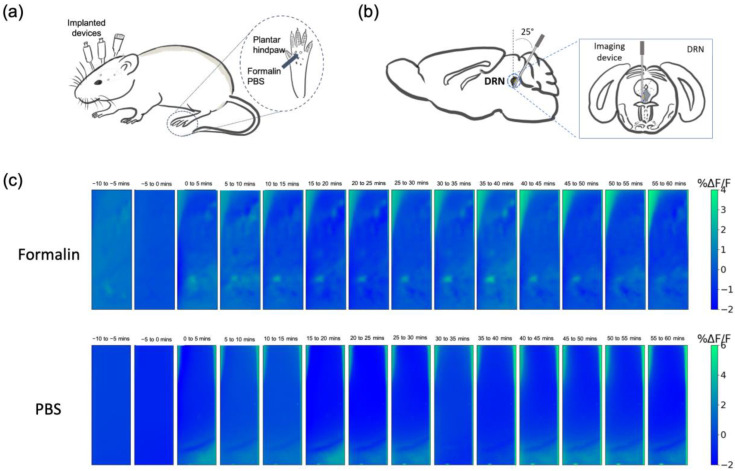
In vivo calcium imaging in the DRN. (**a**) Schematic diagram showing pain stimulation in mouse. Formalin or PBS was injected into the mid plantar of the right mouse’s hind paw. (**b**) The illustration of device implantation in the left DRN. (**c**) Representative fluorescence changes averaged every 5 min in the formalin and the PBS group. The time shown refers to the minutes after the injection.

**Figure 3 ijms-24-06654-f003:**
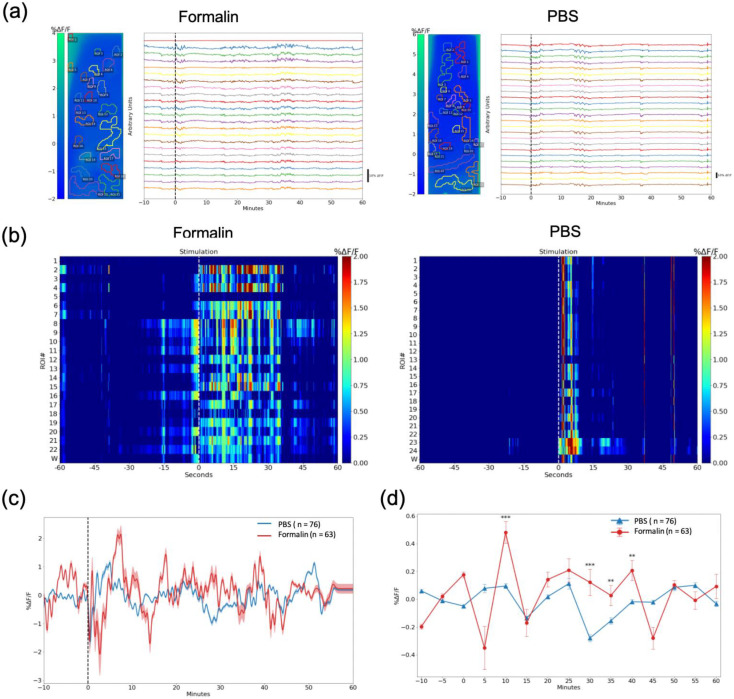
Calcium traces of specific regions of interest (ROIs). (**a**) ROIs in the formalin and the PBS groups; **Left**: ROIs, each color representing each ROI; **Right**: calcium traces for each selected ROIs. The device was implanted in the left DRN. (**b**) Representative heatmap showing fluorescence changes (Δ*F*/*F*_0_) 60 s before and after the injection of formalin and PBS; the number in this figure indicates ROI, and w refers to a whole image. The figure showed the difference in response of the DRN immediately after nociception. (**c**) Average fluorescence changes of G-CaMP6 after the intra-plantar injection of formalin and PBS (60 min). The dashed black line shows the injection (*n* = 63 ROIs in the formalin group, *n* = 76 ROIs in the PBS group). (**d**) Statistical analysis for the fluorescence activity in 5 min time-lapse. Data are presented as mean ± SEM, with the error bars or shaded region indicating the SEM. Statistical analyses were performed using Mann Whitney U tests. ** *p* < 0.01, and *** *p* < 0.001.

**Figure 4 ijms-24-06654-f004:**
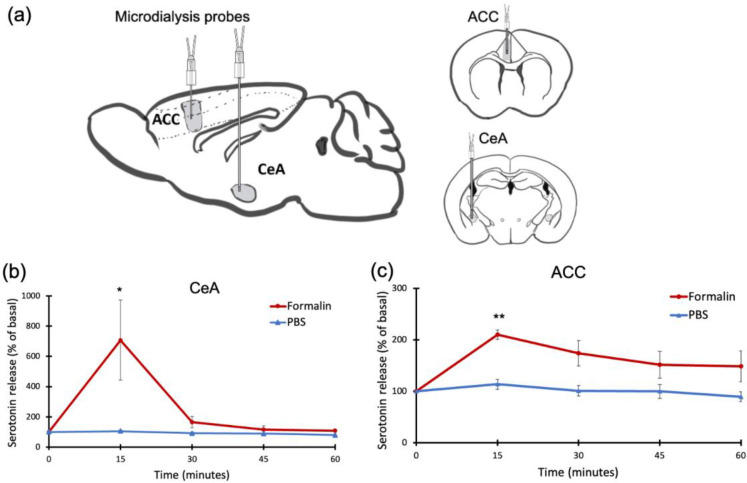
Serotonin concentration in the CeA and the ACC, measured by microdialysis. (**a**) Schematic diagram showing the microdialysis probes in the CeA and the ACC. All the devices were implanted in the left hemisphere. (**b**) Serotonin concentration in the CeA increased 15 min after formalin injection, but not after PBS injection. (**c**) Serotonin concentration in the ACC increased after formalin injection, but not after PBS injection. Data are presented as mean ± SEM, with the error bars indicating the SEM. Statistical analysis was performed using unpaired t-tests between the formalin and PBS groups for each time point (*n* = 5). * *p* < 0.05, ** *p* < 0.01.

**Figure 5 ijms-24-06654-f005:**
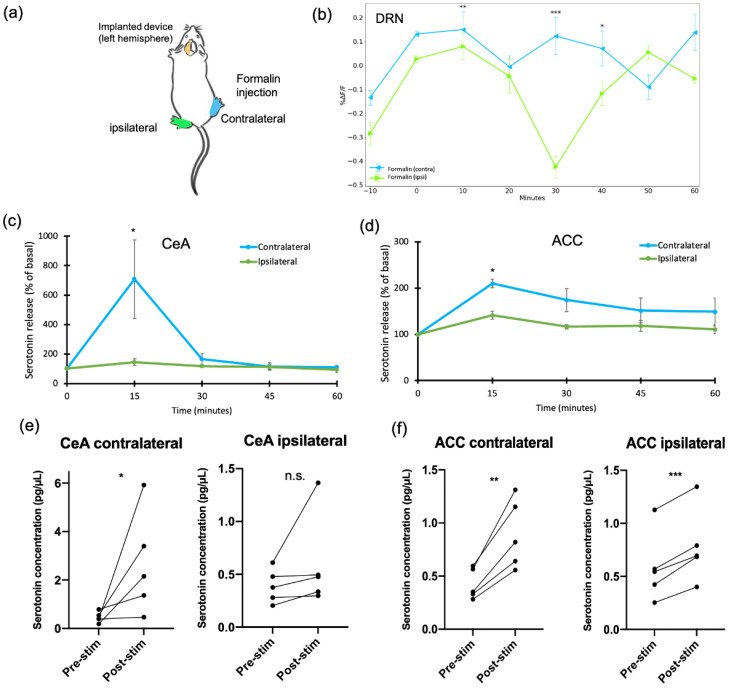
The response of the DRN, the CeA, and the ACC towards formalin injection under contralateral and ipsilateral stimulation. (**a**) The schematic diagram for contralateral or ipsilateral injection of formalin in the mouse mid-plantar hind paw. (**b**) Fluorescence changes (Δ*F*/*F*_0_) of G-CaMP6 in the DRN after formalin stimulation in contralateral and ipsilateral. (**c**) Serotonin release in the CeA comparing contralateral and ipsilateral. (**d**) Serotonin release in the ACC comparing contralateral and ipsilateral. Statistical analysis was performed using unpaired t-tests to compare between contralateral and ipsilateral (*n* = 5). (**e**) Serotonin release in the CeA comparing pre- and post-stimulation. (**f**) Serotonin release in the ACC comparing pre- and post-stimulation. Pre-stimulation was 15 min of sample collection before the formalin injection, and post-stimulation was 15 min of sample collection after the formalin injection. Paired t-tests were performed to compare pre- and post-stimulation (*n* = 5). Data are presented as mean ± SEM, with the error bars indicating the SEM. * *p* < 0.05, ** *p* < 0.01, and *** *p* < 0.001, n.s.: not significant.

**Figure 6 ijms-24-06654-f006:**
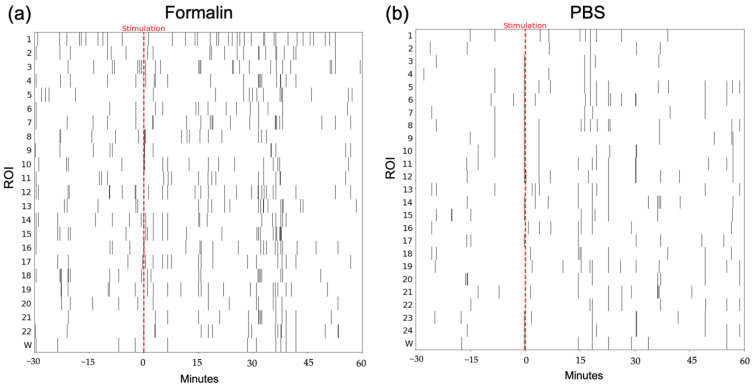
Representative peak raster plot of calcium traces. (**a**) Formalin group. (**b**) PBS group. The red dashed line indicates the stimulation. The *x*-axis shows the time (30 min before and 60 min after the injection), and the *y*-axis shows the ROIs; w refers to the whole frame.

**Figure 7 ijms-24-06654-f007:**
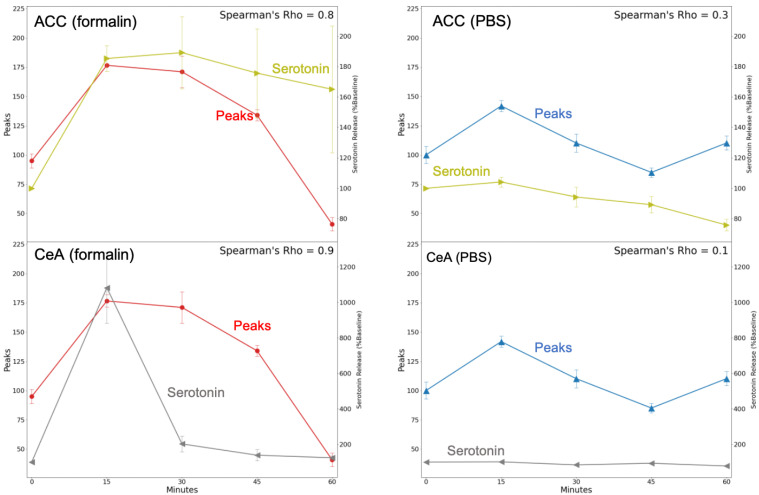
Correlation between calcium imaging peaks of the DRN and serotonin release in the CeA and the ACC using Spearman’s rank correlation test. The red line indicates DRN calcium peaks after formalin injection, the blue line indicates DRN calcium peaks after PBS injection, the gray line indicates serotonin release in the CeA, and the olive indicates serotonin release in the ACC.

**Figure 8 ijms-24-06654-f008:**
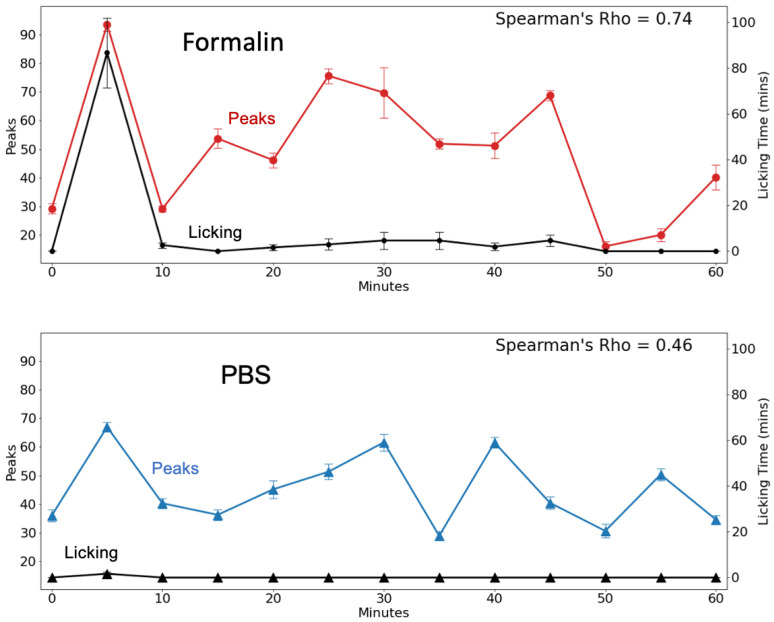
Correlation between calcium imaging of the DRN and licking behavior using Spearman’s rank correlation test. The red line indicates DRN calcium peaks after formalin injection. The blue line indicates DRN calcium peaks after PBS injection. The black line indicates mouse licking behavior.

**Figure 9 ijms-24-06654-f009:**
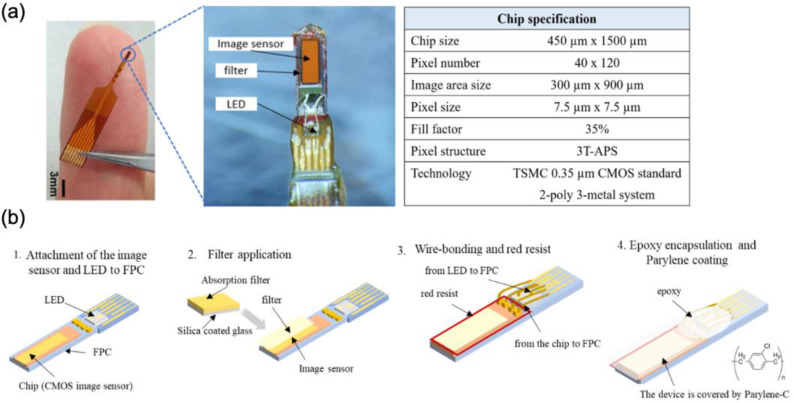
Device Fabrication. (**a**) CMOS implantable device for observing neural activity in the deep brain region, including the device specification. (**b**) Device assembly process.

**Figure 10 ijms-24-06654-f010:**
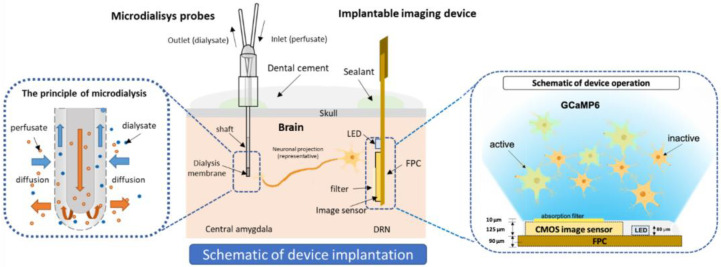
The schematic diagram of microdialysis imaging setup in the brain.

**Figure 11 ijms-24-06654-f011:**
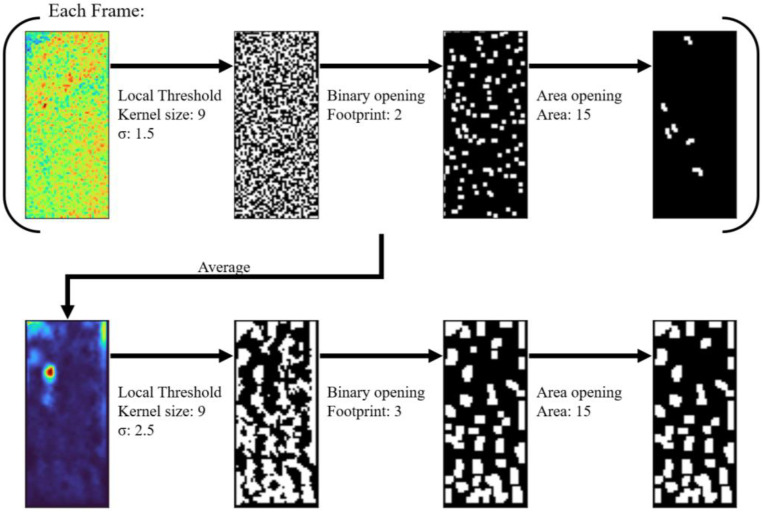
Schematic of ROI selection algorithm. Blue is low, red is high.

## Data Availability

Data is available upon request.

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
