# Peer review of "Multi-Region Microdialysis Imaging Platform Revealed Dorsal Raphe Nucleus Calcium Signaling and Serotonin Dynamics during Nociceptive Pain"

_ijms, 2023, doi:10.3390/ijms24076654_

Round 1

Reviewer 1 Report

The authors of this study developed a multi-region imaging-microdialysis system to investigate neural activity and neurotransmitter release during nociception. Using this system, they measured simultaneously calcium imaging fluorescence to visualized neural activity in the dorsal raphe nucleus, and serotonin release in the central nucleus of the amygdala and the anterior cingulate cortex. Both parameters were measured following nociception induced by formalin injection in the mouse’s hind paw. They found a biphasic increase in calcium fluorescence in response to pain stimulation, the acute and inflammatory phases, and showed that the increase in calcium fluorescence intensity in the dorsal raphe nucleus correlated with elevation in serotonin release in both the central nucleus of the amygdala and the anterior cingulate cortex. Additionally, calcium increase positively correlated with mouse-licking behavior, and was higher for contralateral stimulation. This study reveal the inter-regional serotonergic connection in the brain during nociceptive pain, and provide new insight into the mechanism of pain sensation. It is important and well written paper. I have only few minor comments:

Fig. 6: “y” axis is not described

Supplementary Fig.1, panel b: This panel does not show fluorescence changes in two groups. Thus, it is not clear what does it show.

Supplementary Fig.1, panel c): This panel shows formalin and lidocaine groups which are not mentioned in the text to this panel. Moreover, “y”  axis (ROIs) is not described in left part of this panel.

Supplementary Fig.1, panel d): Figure shows  lidocaine and PFA, not formalin and PBS as stated in the text. Abbreviation ‘G-CaMP6“ is not explained in the text to this figure.

Supplementary Fig.2: Please describe the anatomical structure shown on the bottom of the bright- field picture.

Supplementary Fig.4: Abbreviation BLA is not explained

Reviewer 2 Report

The authors ultralight microimaging device for calcium imaging with microdialysis to simultaneously visualize neural activity in amyagala and anterior cingulate cortex during acute pain stimulation. It addressed the question of live imaging in the brain with mutiple-site recording device. It improved the technique of brain imaging with multiple site implantation. If the recording device could detect both sides of the brain, it will be better.

The sides of the deveice implantation and drug injection were not specifically described in the legends of Fig 1-4 and the corresponding contents in the results, which could be clarified. For example, "imaging device implantation in the left DRN and the microdialysis probes in the left CeA and the ACC."
